# Numerical Simulation Study on Relationship between the Fracture Mechanisms and Residual Membrane Stresses of Metallic Material

**DOI:** 10.3390/jfb13010020

**Published:** 2022-02-21

**Authors:** Yan Yik Lim, Azizi Miskon, Ahmad Mujahid Ahmad Zaidi, Megat Mohamad Hamdan Megat Ahmad, Muhamad Abu Bakar

**Affiliations:** 1Faculty of Defence Science and Technology, National Defence University of Malaysia, Prime Camp, Sungai Besi, Kuala Lumpur 57000, Malaysia; myylim@gmail.com (Y.Y.L.); mujahid@upnm.edu.my (A.M.A.Z.); 2Faculty of Engineering, National Defence University of Malaysia, Prime Camp, Sungai Besi, Kuala Lumpur 57000, Malaysia; megat@upnm.edu.my; 3Faculty of Medicine and Defence Health, National Defence University of Malaysia, Prime Camp, Sungai Besi, Kuala Lumpur 57000, Malaysia; muhamadbakar@upnm.edu.my

**Keywords:** numerical simulation, finite element, cartridge brass, fracture mechanisms, perforation, residual membrane stress

## Abstract

The distribution and dissipation energies in fracture mechanisms were a critical challenge to derive, especially for this ultra-thin sample. The membrane failure, which is the end of the fracture mechanisms, is a result of the cone wave reflections from the backend membrane boundaries. These reflections delay the failure processes due to the shock impacts. To compare these results with the experimental work, a numerical simulation was conducted for these processes. The cylinder-shaped rigid projectile was impacted using a frictionless Lagrange solver. The target was a cartridge brass circle plate clamped at its perimeter, and its zone was refined to a ten-times higher meshing density for better analysis. The erosion and cut-off controls involved a zero-gap interaction condition and an instantaneous geometric erosion strain of 200%. Due to the maximum projectile velocity of 382 m/s having the slowest perforation, the target thickness was found to be 5.5 mm. The fracture mechanism phenomena, such as tensile, compressive, through-thickness, and growth in-plane delamination, propagating delamination, and local punch shear waves were observed. After deducting tensile and flexural strengths from the last experiment, a total residual membrane stress of 650 MPa was found. This result indicated a relationship between the fracture mechanisms and residual membrane stresses of metallic material.

## 1. Introduction

The osteoarthritis patients suffered from decreased cell viability [1] during cartilage tissue cell regeneration therapy [2]. This is due to the fact that most of the articular cartilage scaffolds used for this therapy [3] did not protect external forces or shock impacts from attacking osteoarthritis patients [4] during their daily activities [5]. The biomaterials used in the scaffolds [1] are harder than the cartilage tissue cells, resulting in the cells failing [6] before them. It is crucial to invent a biomaterial that has the requisite mechanical strength [7] and fails before the cells. One of the most well-known standards used for this concept is the body armor compliance tests from the National Institute of Justice (NIJ) [8]. Body armor and biomaterials should be failed first, prior to preventing any blunt-force trauma effect [9,10] or neo-cartilage tissue cell-rupture effect, respectively [11]. Therefore, the ballistic impact test for body armor is a suitable measurement for biomaterial performance [12].

A study on the reduced ballistic limit velocity [12] of graphene membranes by Meng et al. [13] found that a specific penetration energy from the kinetic energy of the projectile was distributed away from the impact zone and propagated to the cone wave development [14]. A relationship was established between the membrane size and the cone wave reflections [15] from the boundaries that induce perforation. In conclusion, the reflected waves had an effect on the sample failure [16] and was greatest for circular free-standing membrane geometries (especially for ultra-thin samples [14,17] such as the biomaterial for articular cartilage scaffold applications in our experimental studies) [18]. Some research had been conducted on the energies distributed and dissipated [19,20,21] during the fracture mechanisms [22,23,24]. Some numerical simulation research had been conducted on the membrane failure mechanisms [14,15,16,25,26,27] to study the distribution and dissipation energies involved at interfaces [21,28]. However, the majority of these studies focused on behaviors and waves [12,22,27,29,30] and not on internal stresses and stress-related defeats [31,32]. Despite the fact that the fracture mechanisms began with the behaviors and waves of delamination processes, the bending at the interfaces and membrane stresses derived from the internal energy of the metallic material were not elucidated.

In the present study, simulation tests on an impact model were conducted to analyze the ballistic limit velocity phenomena and fracture mechanisms of the cart brass target plate. The target plate was meshed using the finite element method, coupled with local and global conditions. The controls, such as material models, solutions, and outputs, were applied to the numerical simulation modelling [24] in order to obtain better simulation results. The ballistic limit thickness of the target plate was determine and used to compare out results with our previous experimental work [18]. This study aims to investigate the relationship between the energies and stresses during the fracture mechanisms.

## 2. Materials and Methods

### 2.1. Material Models

The important properties of material models for three materials, such as cartridge (cart) brass [33], lead [34], and copper [35], were used in the numerical simulation as shown in Table 1. For the other properties of material models, please refer to Table A1 of Appendix A. The material models of Johnson–Cook [36], Steinberg–Guinan, and Piecewise–Johnson–Cook for cart brass, lead, and copper, respectively, were selected from the library of Ansys Autodyn non-linear dynamics analysis software (Autodyn, version 13, Canonsburg, WA, USA).

### 2.2. Geometry Parts of Projectile and Target Plate

The bullet head was drawn from the actual bullet in the middle, which was changed from the round-nosed tip to a blunt-nosed tip [37], as shown on the left and right of Figure 1, respectively. This tip parameter was neglected since there was no air resistance environment in the simulation. This s common practice in geometry meshing in order to reduce the chances of simulation termination due to lots of energy loss. Therefore, the projectile was simplified into the 40 mm length of cylinder-shaped projectile geometry parts with dimensions of 10 mm and 5 mm for the outer and inner diameters, respectively, as shown at the right of Figure 1.

The projectile size was scaled up two times in order to obtain higher quality of meshing. Therefore, the size of the target was also scaled up two times. This target was impacted by the projectile from the x-direction, as shown by the red double-head arrow in Figure 2. The target part for this simulation was a thin cart brass circle plate with a 60 mm radius and a 5.5 mm thickness, as shown in Figure 3.

### 2.3. Finite Element Meshing Method

As shown in Figure 3, the original point 0, I-direction, and J-direction were defined at the bottom center of the target plate, opposite x-axial and y-axial, respectively. The original element dimensions were 0.25 mm thick × 0.2 mm^2^ in area. After the finite element meshing, the area of the projectile impacted on the target was refined to a ten-times higher density than others [38], as shown by the yellow circle in Figure 3. This meant that a single element was divided into ten elements in the J-direction. This refined the lower part of the plate from one to six in the J-direction. In total, 86,428 elements were used for the refined impact zone of the target in this finite element meshing process in order to analyze it more explicitly.

## 3. Simulation Tests and Results

### 3.1. Local and Global Conditions

The local condition of the initial velocity in x-direction was defined as 364 m/s and complied with the NIJ IIA [8]. The global conditions, such as boundary and gap interaction, were selected before being converted to three-dimensional geometries [39]. The clamp boundary condition was applied to J-line index of 61, which was located at the top of circle plate perimeter (as shown in Figure 4). This clamp was defined as the velocity constant of zero at x-axial and y-axial. The Lagrange solver, coupled with erosion and interaction without friction and a stiffness-based hourglass control, was used for the direct impact from the projectile part to the target part [40]. In order to save simulation time, the gap between the projectile and the target plate was minimized. The interaction boundary condition of zoning was applied to this gap by using the translate function in order to expand a transformation gap with a non-friction zone.

### 3.2. Material Model Controls

The projectile materials, such as copper and lead, were defined as rigid bodies, resulting in no failure or erosion. Therefore, there was no control applied. The important controls of the material model, such as erosions and cut-offs, were applied to cart brass to minimize the computational effort, as shown in Table 2 and Table 3 [33,34,35]. For the other erosion and cut-off controls, please refer to Table A2 of Appendix A. In order to reduce errors during the erosion process and failure mechanisms, it was critical to define the geometric erosion strain as 2.0 (200%) and instantaneous [41]. In order to investigate the details of failure mechanism progressions, the material model cut-off control factors such as minimum density and maximum expansion were defined to be 0.0001 and 0.1, respectively. For the smoothed-particle hydrodynamic progressions, the minimum and maximum density factors were defined largely due to the fact that there was no failure mechanism.

### 3.3. Three-Dimensional Geometry Conversion and Solution Controls

The two-dimensional axial and x-axial symmetries must be predefined in order for the model to convert from two-dimensional geometries to three-dimensional geometries at the x-axial symmetry. The three-dimensional geometries with the respective coordination and dimensions were loaded with the conditions of clamp, gap interaction, and initial velocity (as shown in Figure 5). The whole numerical simulation process can be checked by clicking the viewing plot on the icon bar. Furthermore, the solution controls, such as wrap-up criteria, time-step options, and solver options, were used to gain the higher quality images during the metallic material failure point.

### 3.4. Simulation Test Results

Autodyn with a finite element code using four-nodes 2D asymmetric element in one integration point was used to analyze the simulation tests explicitly. The Lagrange framework model with visco-plasticity and ductile damage computational mode was coupled with non-friction interaction and stiffness-based hourglass control for the element-kill algorithm to minimize the error in the erosion process and maximize the efficiency. In order to simplify these simulation test studies, the ballistic impacts were limited to the perpendicular direction of the target based on the assumption that the maximum impacts occurred [41]. Therefore, the other impact directions were neglected for these simulation test studies. The simulation tests were run and exhibited at the cycles of 0, 10,415, 21,664, and 38,969 for the times of 0, 0.05, 0.1 and 0.15 ms, respectively (as shown in Figure 6). Various velocities of NIJ IIA were inputted into the simulation tests in order to obtain a ballistic limit velocity (BLV) that just perforated the target (as shown at the bottom right of Figure 6).

## 4. Simulation Results Analyses

### 4.1. Determination of the Ballistic Limit Velocity Phenomenon

BLV [14] was defined as the projectile velocity that penetrated the target with at least 50% reliability or kinetic energy loss [8]. This was similar to the simulation requirement to determine the BLV as the projectile velocity that perforated the cart brass target plate at the slowest rate (as shown in Figure 7a). The numerical simulation was described by the projectile head impacting target phenomena of BLV, completely perforating velocity, almost perforating velocity, and nearly perforating velocity (as shown in Figure 7a–d, respectively). For the almost and nearly perforating velocity phenomena, the projectile velocity penetrated the target at different depths but was unable to perforate the target.

### 4.2. Determination of Ballistic Limit Thickness of Target Plate

The minimum and maximum initial projectile velocities of 364 m/s and 382 m/s, respectively, were complied with NIJ IIA and inputted into the tests [8], as shown at the bottom of Figure 6. The BLV was determined through ten simulation tests with both initial velocities impacted on various cart brass target plate thickness (as shown in Figure 7a). The BLV was only obtained in the test while the minimum initial projectile velocity and maximum initial projectile velocity showed penetration and perforation of the target, respectively. This result was found in test number 10 at 5.5 mm target plate thickness where the BLV occurred (as shown in Figure 8).

### 4.3. Fracture Mechanism Analysis

While the projectile impacted the target plate, the tensile and compressive strain waves were developed and reacted [12] from the y-axial and spherical x-axial, respectively, as shown at the left of Figure 9. The projectile continued to progress in the x-axial to counter the through-thickness wave in the opposite direction [25], resulting in a transverse shear force. This force continued to form a wave to counter the projectile progression, resulting in the growth in-plane delamination wave [42]. This phenomenon especially occurred at the first few predefined interfaces or the refined zone of the cart brass target plate which was elucidated by selecting the simulation cycles of 10,415 and 21,664 (as shown in Figure 9). This enabled us to have a more insightful evaluation of the fracture mechanism [22] related to the delamination growth in the representative volume cell at each material integration point [22]. The projectile further progressed to form a pyramid shape in the interfaces due to the reaction from the propagating delamination wave, as shown at the top right of Figure 9. All of these reaction waves continued to counter the projectile progression, following with a larger deformation from all interfaces until a cone shape was formed at the back of the target. This was the local punch shear-cone wave that bulged dynamically from the back-face of the target. Lastly, the target failed by the projectile breaking the membrane tensile failure strain.

### 4.4. Microstructure Analysis

The BLV was defined as a projectile that just had enough energy to break the cart brass target plate [8]. As a result, all of the projectile kinetic energy was transferred [14] into the cart brass internal energy [27]. The constant energy and projectile mass equations have been recorded in Equations (1) and (2), respectively. As shown in Equation (3), the internal stress of the cart brass [31,32] was calculated by inserting Equation (2) into Equation (1).
(1)$$\mathrm{Projectile}\;\mathrm{Kinetic}\;\mathrm{Energy}\;\frac{1}{2}m_{p}v^{2}=\sigma_{CB}Ah\;\mathrm{Cart}\;\mathrm{Brass}\;\mathrm{Internal}\;\mathrm{Energy}$$
(2)$$\mathrm{Projectile}\;\mathrm{Mass}\;m_{P}=\rho_{C}\frac{\pi }{4}D_{C}{}^{2}L+\rho_{L}\frac{\pi }{4}D_{L}{}^{2}L$$
(3)$$\mathrm{Cart}\;\mathrm{Brass}\;\mathrm{Stress}\;\sigma_{CB}=\frac{\frac{\pi }{8}Lv^{2}\left(\rho_{C}D_{C}{}^{2}+\rho_{L}D_{L}{}^{2}\right)}{\frac{\pi }{4}D_{CB}{}^{2}h}$$
where *ρ_c_* and *ρ_L_* are the copper and lead densities of 8.9 and 11.34 g/cm^3^, respectively; *D_C_*, *D_L_*_,_ and *D_CB_* are the copper, lead and cart brass diameters of 10, 5 and 10 mm, respectively; and *L*, *h*, and *v* are the projectile length, the cart brass target plate thickness, and the maximum projectile velocity of 40 mm, 5.5 mm, and 382 m/s, respectively.

All of the property values were inserted into Equation (3) in order to obtain the total cart brass stress, *σ_CB_* of 6.227 × 10^9^ Nm^2^ or 6227 MPa.

### 4.5. Prediction of Residual Membrane Stress

As shown in Figure 9, the fracture mechanism consisted of waves of tensile, compressive, through-thickness, growth in-plane delamination, propagating delamination, and local punch shear [22]. All of the waves were caused by the energy dissipation in the cart brass target plate [43], resulting in the failure stress of the target plate [14]. All of these stresses were needed to break the cart brass target plate, which was equal to 6227 MPa.

In order to compare our results to our previous experimental work, two assumptions were made on a product made of 30% cart brass filler and 70% polyester urethane liquid (PEU). These assumptions were that the total cart brass stress received significant contributions by the 30% of cart brass filler (but not the PEU), and that the bio-fabrication process did not contribute any significant stress. As a result, the simulation product with 30% cart brass filler had a total brass stress of 1868 MPa (as shown in Table 4).

For the last experimental work, a biomaterial was fabricated using the metallic filler polymer reinforced method [44,45,46] to provide the requisite mechanical and structural strengths [47] for the scaffold integrity [48]. The tensile strength and flexural strength for this biomaterial made of 30% low brass filler and 70% polyester urethane liquid were found to be 203 MPa and 1015 MPa, respectively [18,49]. Although there was a different nominal composition between the cart brass (70% copper and 30% zinc) and low brass (80% copper and 20% zinc), the maximum tensile and yield strengths for both brass products were the same [50]. Therefore, another assumption of having similar mechanical strengths for the cart brass and low brass was made. For the details of cart brass and low brass properties, please refer to Table A3 of Appendix A. As shown in Table 4, the total residual membrane stress was equal to the total brass stress after deducting tensile strength and flexural strength, which was equal to 650 MPa. Based on the fracture mechanism stresses, this total residual membrane stress was unknown and implicated the compressive, through-thickness, and local punch shear stresses in the overall process.

## 5. Conclusions

The cart brass target plate result was at a 5.5 mm thickness, wherein the phenomenon was obtained by the maximum projectile velocity of 382 m/s perforated at the slowest rate. The fracture mechanisms which consisted of waves of tensile, compressive, through-thickness, growth in-plane delamination, propagation delamination, and local punch shear were observed. These waves were the energy dissipation that occurred on the cart brass target plate before failure. By calculating all of the projectile kinetic energy that was transferred into cart brass internal energy, the total cart brass stress, *σ_CB_* of 6227 MPa, was obtained. By using 30% of the total cart brass stress, a total brass stress of 1868 MPa was calculated to compare with the last experimental work. The total residual membrane stress of 650 MPa was calculated by deducting the total brass stress with tensile strength (203 MPa) and flexural strength (1015 MPa). This total residual membrane stress implicates the compressive, through-thickness, and local punch shear stresses in the process.

This numerical simulation finding was successfully compared to the mechanical strength results from experimental work. The relationship between the tensile and flexural strengths and the total brass stress was also successfully established. Furthermore, the relationship between the fracture mechanisms and the residual membrane stresses was proposed in this study. This provides an important guideline for future experimental work on membrane failure stresses. The residual membrane stresses during the fracture mechanisms are crucial to understand, elucidate, and clarify in future experimental work in order to develop better performance of the biomaterials. This biomaterial with higher stresses is used in articular cartilage scaffold, which aids in absorbing and delaying external forces and shocks that attack osteoarthritic patients during their daily activities [5]. This is an innovative feature that has the potential to be developed for the prevention of blunt-force trauma effects [9].

## Figures and Tables

**Figure 1 jfb-13-00020-f001:**
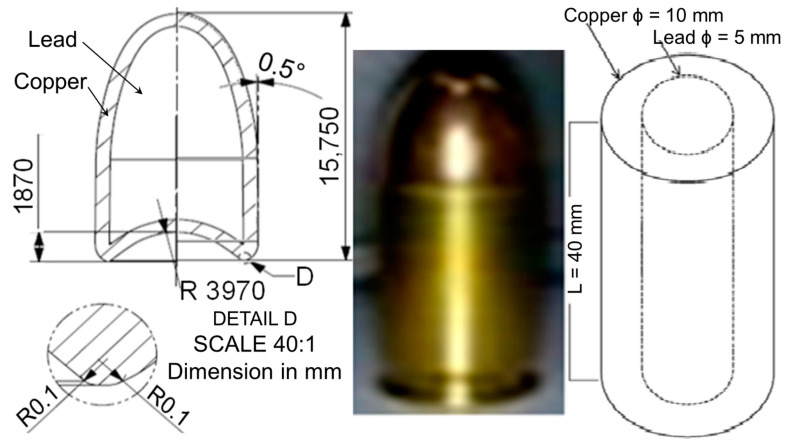
Geometry of bullet head (**Left**), actual bullet and projectile (**Right**). Note: the actual bullet dimensions were 9 mm diameter × 19 mm length full metal jacket round-nosed bullet, as specified by the NIJ Standard Type Threat Level IIA (NIJ IIA) [8].

**Figure 2 jfb-13-00020-f002:**
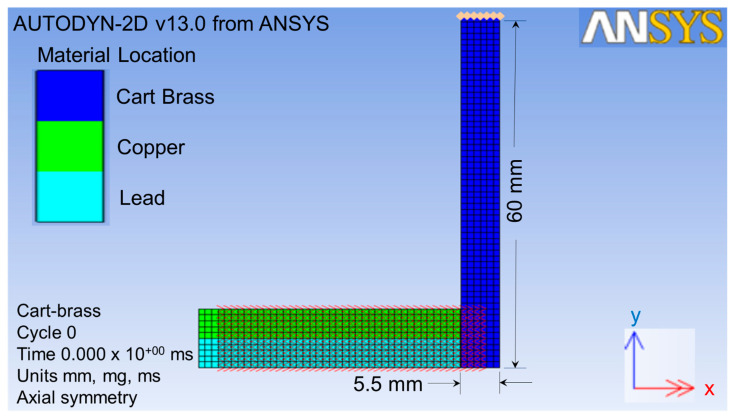
Geometry parts of the projectile and target with the impact direction.

**Figure 3 jfb-13-00020-f003:**
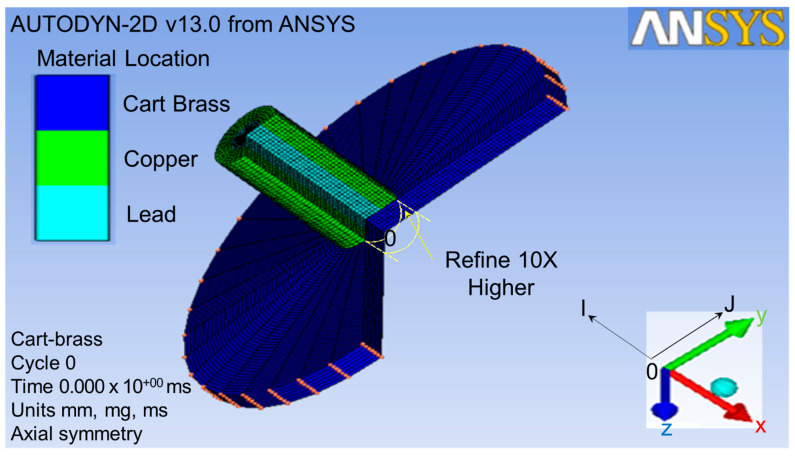
Finite element meshing for geometry parts.

**Figure 4 jfb-13-00020-f004:**
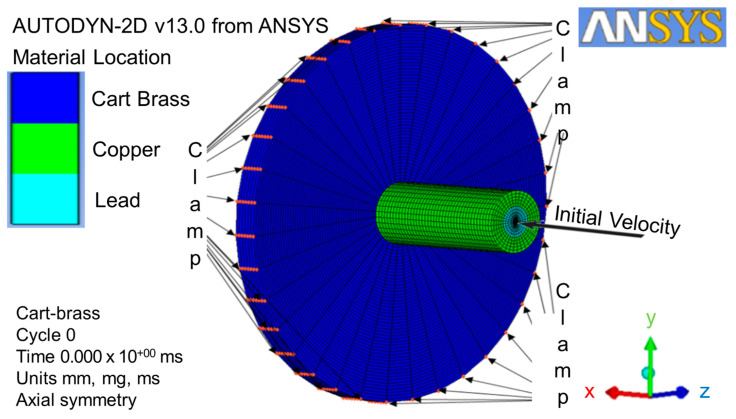
Initial velocity local condition, clamp, and gap interaction global conditions.

**Figure 5 jfb-13-00020-f005:**
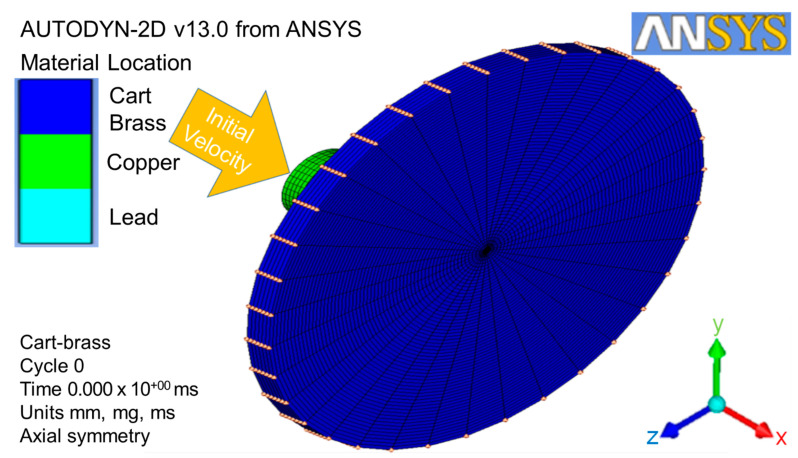
Three-dimensional geometry conversion with conditions of clamp, interaction, and initial velocity.

**Figure 6 jfb-13-00020-f006:**
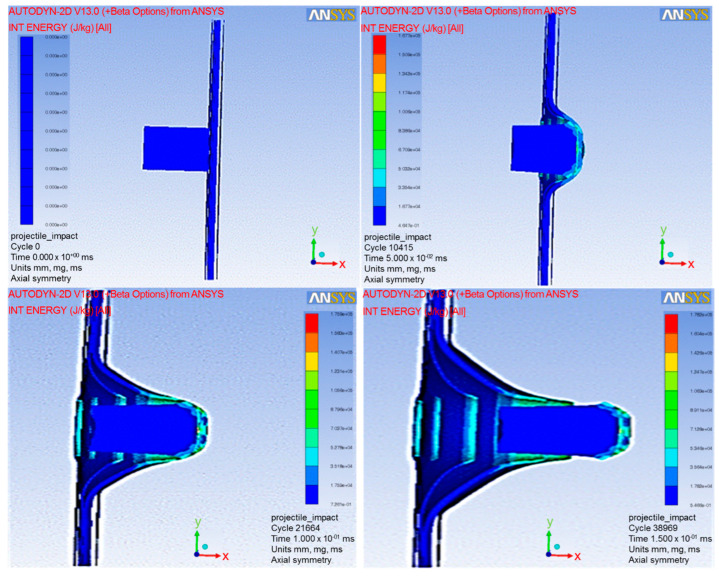
Autodyn simulation test running at various cycles and times.

**Figure 7 jfb-13-00020-f007:**
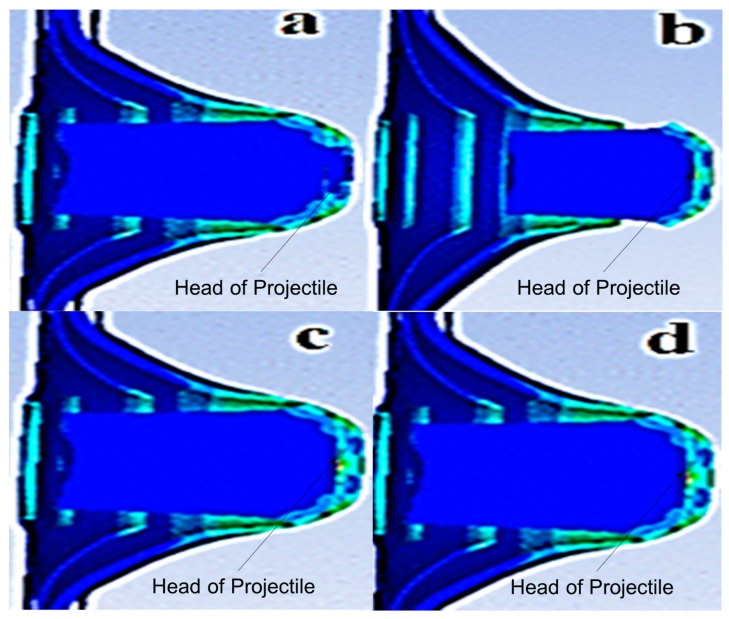
Numerical simulation impact phenomena of (**a**) ballistic limit velocity; (**b**) completely perforating velocity; (**c**) almost perforating velocity; and (**d**) nearly perforating velocity.

**Figure 8 jfb-13-00020-f008:**
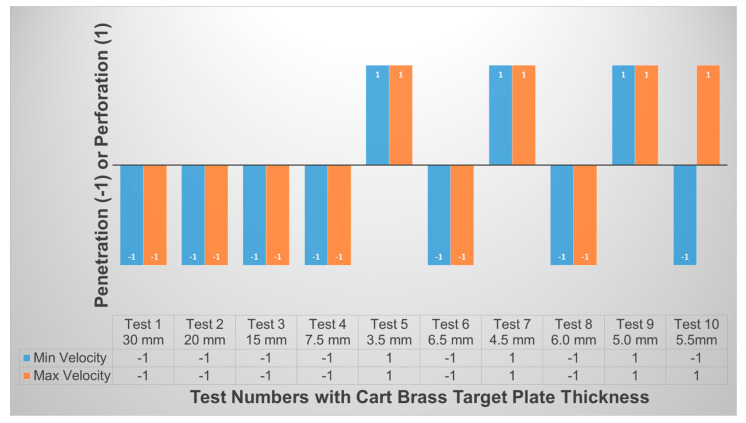
Penetration and perforation of minimum and maximum initial projectile velocities impacted on various cart brass target plate thicknesses.

**Figure 9 jfb-13-00020-f009:**
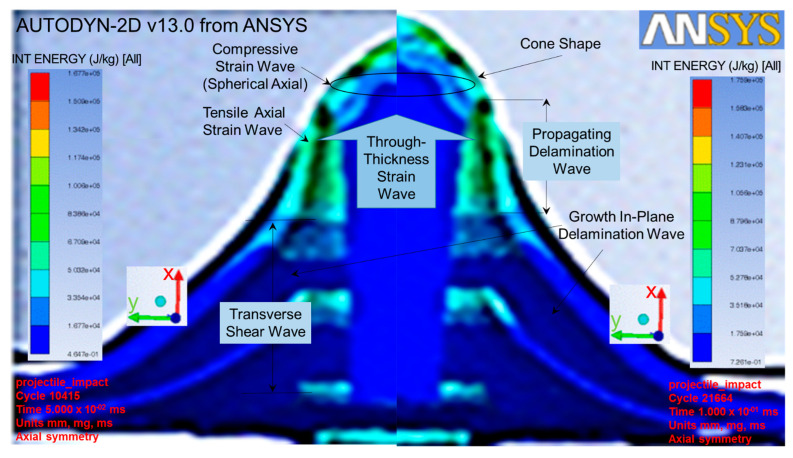
Fracture mechanisms occurred in cart brass target plate while projectile impacted at simulation cycles 10415 (**Left**) and 21664 (**Right**).

**Table 1 jfb-13-00020-t001:** Important properties of cartridge (cart) brass, lead, and copper material models.

Material	Cart Brass	Lead	Copper
Model	Johnson Cook	Steinberg Guinan	Piecewise Johnson Cook
Reference density, g/cm^3^	8.45	11.34	8.9
Gruneisen coefficient	2.04	2.74	2.0
Shear Modulus, GPa	37.4	8.6	46.4
Yield Stress, MPa	112	8	120

**Table 2 jfb-13-00020-t002:** Important erosion controls of material model.

Erosion Control	Strain	Type
Geometric Strain	2.0	Instantaneous

**Table 3 jfb-13-00020-t003:** Important cutoff controls of material model.

Cutoff Control	Minimum	Maximum
Density or Expansion Factor	0.0001	0.1
Density Factor for Smoothed-Particle Hydrodynamics	0.2	3.0

**Table 4 jfb-13-00020-t004:** Summary of stress and strength for cart brass and low brass filler percentages.

Brass Filler Percentage	Total Brass Stress, MPa	Tensile Strength, MPa	Flexural Strength, MPa
30% cart brass	1868		
30% low brass		203	1015
Total Residual Membrane Stress	650		

## Data Availability

Not applicable.

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
