# Peer review of "Numerical Simulation Study on Relationship between the Fracture Mechanisms and Residual Membrane Stresses of Metallic Material"

_jfb, 2022, doi:10.3390/jfb13010020_

Round 1

Reviewer 1 Report

This paper presents the numerical simulation study on relationship between the Fac-2 ture mechanisms and residual membrane stresses of metallic 3 material. However, the paper is not well organized. The contribution of the paper is not highlighted. The Introduction and the Conclusions are not well organized. In addition, there is a lack of innovation in the modelling, solution method and results. It can not be accepted for publicaiton as this format.

Reviewer 3 Report

Comments to the Author

Title: Numerical Simulation Study on Relationship between the Fracture Mechanisms and Residual Membrane Stresses of Metallic Material

Authors: Lim Yan Yik; Azizi Miskon; Ahmad Mujahid Ahmad Zaidi; Megat Mohamad Hamdan Megat Ahmad and Muhamad Abu Bakar

Comments: The distribution and dissipation energies in the fracture mechanisms were a critical challenge to derive, especially for this ultra-thin sample. It has been shown that the fracture mechanisms began with the behaviors and waves of delamination processes. However, the membrane stresses derived from the internal energy of the metallic material were not elucidated. The authors presented a detailed numerical simulation work to investigate the energies and stresses that occurred in the fracture mechanisms. The simulated results show that the fracture mechanisms that consisted of waves of tensile, compressive, through-thickness, growth in-plane delamination, propagation delamination and local punch shear were observed, and demonstrate the relationship between the fracture mechanisms and residual membrane stresses of metallic material. All the simulation results are well interpreted and the conclusion is convincing. I therefore recommend to publish after my concerns are properly addressed.

  1. In the Introduction section, the author does not explain the background of this article well. It is recommended that this section be divided into the background of the study, the current status of the study and the purpose of the author's study.
  2. The fracture mechanism and microstructure analyses can be further deepened by comparing the present numerical simulation work with the experimental reports or with other numerical simulation works.
  3. The author can be represented in a more standard icon form in Figure 8 and can explain the meaning of “1” and “-1” in the figure in detail. Also, the Figure 6, 7 and 9 is not clear.
  4. The latest papers of metallic membrane and stress analysis are suggested for citation which may be helpful for present manuscript: ACS Appl. Mater. Interfaces 2021, 13, 55712−55725; Int. J. Plast. 142 (2021) 102997.

Round 2

Reviewer 1 Report

The paper has been carefully revised according to the reviewers' comments, it can be accepted for publication.

Reviewer 2 Report

I studied the revised manuscript and I accepte the author's answers to our suggestions and comments.

According to my opinion, the revised manuscript

jfb-1584551-peer-review-v2

is now suitable for publication in the JFB.